

# Biodiversity of the Great Barrier Reef—how adequately is it protected?

Zoe T. Richards[1,2] and Jon C. Day[3]

[1] Trace and Environmental DNA Laboratory, School of Molecular and Life Sciences, Curtin University of Technology, Perth, WA, Australia
[2] Aquatic Zoology Department, Western Australian Museum, Welshpool, WA, Australia
[3] ARC Centre of Excellence for Coral Reef Studies, James Cook University of North Queensland, Townsville, QLD, Australia

Corresponding author
Zoe T. Richards,
zoe.richards@curtin.edu.au

## ABSTRACT

**Background:** The Great Barrier Reef (GBR) is the world's most iconic coral reef ecosystem, recognised internationally as a World Heritage Area of outstanding significance. Safeguarding the biodiversity of this universally important reef is a core legislative objective; however, ongoing cumulative impacts including widespread coral bleaching and other detrimental impacts have heightened conservation concerns for the future of the GBR.

**Methods:** Here we review the literature to report on processes threatening species on the GBR, the status of marine biodiversity, and evaluate the extent of species-level monitoring and reporting. We assess how many species are listed as threatened at a global scale and explore whether these same species are protected under national threatened species legislation. We conclude this review by providing future directions for protecting potentially endangered elements of biodiversity within the GBR.

**Results:** Most of the threats identified to be harming the diversity of marine life on the GBR over the last two–three decades remain to be effectively addressed and many are worsening. The inherent resilience of this globally significant coral reef ecosystem has been seriously compromised and various elements of the biological diversity for which it is renowned may be at risk of silent extinction. We show at least 136 of the 12,000+ animal species known to occur on the GBR (approximately 20% of the 700 species assessed by the IUCN) occur in elevated categories of threat (*Critically Endangered, Endangered* or *Vulnerable*) at a global scale. Despite the wider background level of threat for these 136 species, only 23 of them are listed as threatened under regional or national legislation.

**Discussion:** To adequately protect the biodiversity values of the GBR, it may be necessary to conduct further targeted species-level monitoring and reporting to complement ecosystem management approaches. Conducting a vigorous value of information analysis would provide the opportunity to evaluate what new and targeted information is necessary to support dynamic management and to safeguard both species and the ecosystem as a whole. Such an analysis would help decision-makers determine if further comprehensive biodiversity surveys are needed, especially for those species recognised to be facing elevated background levels of threat. If further monitoring is undertaken, it will be important to ensure it aligns with and informs the GBRMPA Outlook five-year reporting schedule. The potential also exists to incorporate new environmental DNA technologies into routine

monitoring to deliver high-resolution species data and identify indicator species that are cursors of specific disturbances. Unless more targeted action is taken to safeguard biodiversity, we may fail to pass onto future generations many of the values that comprise what is universally regarded as the world's most iconic coral reef ecosystem.

## INTRODUCTION

The Great Barrier Reef (GBR) is a diverse ecosystem extending for more than 2,300 km along Australia's northeast coast. It is recognised internationally as being of outstanding universal value (*United Nations Educational, Scientific and Cultural Organisation, 1981*; *Lucas et al., 1997*; *GBRMPA, 2014a*). Its diversity includes but is not restricted to over 410 species of hard coral, over 1,620 species of fish, 2,000 species of sponge, 14 species of sea snake, six of the world's seven species of marine turtle, at least 300 mollusc species, 630 species of echinoderm, and 500 species of marine alga. No other World Heritage Area on the planet contains such diversity (*Day, 2016*) and the GBR's exceptional biodiversity values were specifically mentioned in two of the four criteria for natural heritage (ix and x) when it was World Heritage listed in 1981 (*United Nations Educational, Scientific and Cultural Organisation, 1981*).

The long-term protection and conservation of biodiversity values is at the core of the primary legislative objective for the GBR (s. 2A of the Act, *Great Barrier Reef Marine Park Act, 1975*) of all recent GBR planning and management documents such as the 2014 Outlook Report (*GBRMPA, 2014a*), the 2014 Strategic Assessment (*GBRMPA, 2014b*) and the Reef 2050 Sustainability Plan (Reef 2050 Plan) (*Commonwealth of Australia, 2015*) reflect this. However, a recent report on the feasibility of the Reef 2050 Plan suggests that maintaining the GBR's Outstanding Universal Value and biodiversity as we know it may no longer be realistic, and it is recommended that the key managerial focus should be preserving the ecological function of GBR ecosystems (*Roth et al., 2017*).

This recommendation is based on the ongoing habitat deterioration of the GBR, even in sectors where there is minimal human influence (*Hughes et al., 2017*). It is also based on the finding that windows of opportunity for recovery after disturbances are narrowing (*Hughes et al., 2018*)—a consequence of the lack of progress that has been made towards global emission and local water quality targets. While such a policy change may be inevitable, it should not preclude biodiversity conservation. In addition to the moral and aesthetic reasons for protecting biodiversity, the conservation of diversity is important for protecting the functioning and resilience of the ecosystem as a whole. This is because inherent variation in species responses to, and recovery after disturbances provides ecological insurance in the face of change (*Nyström et al., 2008*). Furthermore, diversity buffers ecosystems against environmental change and increases the stability

and functioning of ecosystem processes (*Griffin et al., 2009*; *Loreau & de Mazancourt, 2013*). Conversely, species loss can accelerate change in ecosystem processes (*Stachowicz, Bruno & Duffy, 2007*; *Perrings et al., 2011*; *Hooper et al., 2012*) and the extinction of a species may have unforeseen impacts (*Dulvy, Sadovy & Reynolds, 2003*).

There is little doubt that quantitative scenarios for the future of biodiversity in the 21st century are bleak (*Pereira et al., 2010*), however there is still a chance to intervene through better policies. Given biodiversity is so intimately related to ecosystem functioning, in this review, we focus on the biodiversity values of the GBR. We acknowledge that in complex and socio-economically valuable ecosystems where there are multiple stakeholders and users, conservation decisions should never be made with biodiversity data alone. However, the purpose of this review is to introduce the biodiversity values of the GBR, summarise the threats directly or indirectly impacting these values, evaluate the status of marine life and examine the influence of scale on threatened species management approaches. Our objective is not to provide an exhaustive commentary on the biodiversity conservation literature (see *Yoccoz, Nichols & Boulinier, 2001*; *Stem et al., 2005*; *Ferraro & Pattanayak, 2006* for general reviews of monitoring and evaluation in conservation), or the challenges faced by environmental managers and policymakers (see *Anthony, 2016*), but rather to highlight opportunities for optimizing the evaluation and protection of biodiversity on the GBR.

## SURVEY METHODOLOGY

The 2014 GBR Outlook Report (*GBRMPA, 2014a*) was chosen as the initial primary source of key references of the key threats/pressures impacting upon the GBR. The 2014 Outlook Report is a comprehensive compilation of information about the GBR underpinned by many references. Searches of the Outlook Report were undertaken against the following keywords: climate change, water quality, coastal development, shipping, unsustainable fishing, diseases, pests, and marine debris. Eighty-two articles that referred to the consequences, impacts or implications of the threats/pressures for species on the GBR were retained and included in Table S1. Table S1 was further augmented by conducting the same keyword searches of the James Cook University library, with only those articles directly relevant to species on the GBR retained. Articles that did not refer to consequences, impacts or implications of the threats/pressures for species were excluded. Precedence was given to articles published within the last 15 years; however, five articles published in the 1990s were included in Table S1 as they provided critical information not available elsewhere. In total, 125 titles were included in our review of the key threats/pressures impacting species within the GBR and these are listed in Table S1. We also conducted Google Scholar searches focusing on the ecological, environmental management, and conservation planning literature for articles relating to the challenges presented by monitoring and managing diversity in coral reef ecosystems, coral reef surrogates and proxy metrics. We examined the 2014 GBR Outlook Report (*GBRMPA, 2014a*) to quantify how many species were known from the GBR and the threatened status of these fauna based on national (EPBC Act: http://www.environment.gov.au/marine/marine-species/marine-species-list) and international

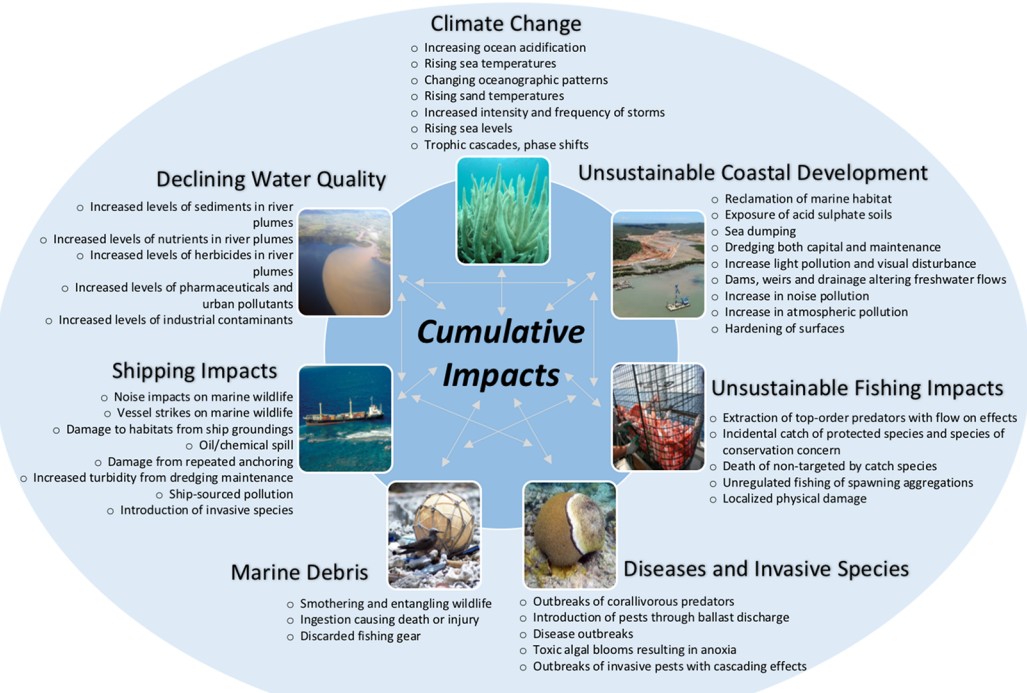

**Figure 1 Cumulative impacts on the Great Barrier Reef.** Cumulative impacts on the Great Barrier Reef that are directly or indirectly affecting species. For more information on key threats/pressures and the implications for species see Table S1. Water quality image—Photographer: C. Honchin, Copyright Commonwealth of Australia (GBRMPA); Marine debris image—Photographer: S. Whiting, Copyright Commonwealth of Australia (GBRMPA); Shipping image—Photographer: P. Howorth, Copyright Commonwealth of Australia (GBRMPA); Coastal Development image courtesy Queensland Government Department of Heritage and Protection under Creative Commons Attribution 3.0 Australia (CC BY) license (http://www.ehp.qld.gov.au/legal/copyright.html). All other images by Zoe Richards.

threatened species lists (IUCN red-list http://www.iucnredlist.org/). This comparison was undertaken for approximately 700 species (including hard corals, sea cucumbers, giant clams, grouper, wrasse, parrotfish, sharks, rays, sea snakes, turtles, whales, dolphins, and dugong) whose conservation status has been assessed at a global level. Lastly we used a Google Scholar search to locate titles that relate to optimal coral reef monitoring and alternative coral reef monitoring technologies.

## RESULTS AND DISCUSSION

### Current status of the diversity of marine life on the GBR

Despite significant management actions (*Fernandes et al., 2005*; *Commonwealth of Australia, 2015*), the diversity of marine life on the GBR is threatened by seven key pressures (climate change, declining water quality, coastal development, shipping impacts, fishing impacts, diseases and pest species, and marine debris) (Fig. 1). These direct or indirect pressures have led to at least 43 impacts which may adversely impact species on the GBR (Fig. 1, see also Table S1). The latest extensive impact, the back-to back thermal stress events in 2015–2016, led to 91% of surveyed GBR reefs experiencing coral bleaching

(*Hughes et al., 2017*). This large-scale coral bleaching event heightens concerns about whether the biodiversity values of the GBR are intact because it occurred at a time when parts of the GBR system were recovering from three category five cyclones (2010–2015, *Perry et al., 2014*), and a legacy of cumulative pressures (e.g. the commercial harvesting of dugongs, green turtles, pearl shell, trochus, and sea cucumbers; catchment management issues, including land clearing, changes to natural water flows and water pollution (*GBRMPA, 2013a*; *Anthony, 2016*)).

Prior to the 2016 bleaching event, both habitats and species in the southern two-thirds of the GBR, and particularly in the inshore areas, had reportedly declined (*Brodie & Waterhouse, 2012*; *GBRMPA, 2009*, *2014a*; *Osborne et al., 2011*). A temporal analysis of the level of coral cover indicated that from 1985 to 2012 reef-wide coral cover declined by an average of 0.53% per year from 28.0% to 13.8% (*De'ath et al., 2012*). Between 1988 and 2003 coral calcification rates are also reported to have declined from 1.96 g cm$^{-2}$ year$^{-1}$ (±0.05) to 1.59 cm$^{-2}$ year$^{-1}$ (±0.04), equivalent to a decline of 14–21% over this period (*Cooper et al., 2008*; *De'ath Lough & Fabricius, 2009*). Degradation was not, however, restricted to the corals; sea grass health, particularly in central GBR was poor (*Fairweather, McKenna & Rasheed, 2011*; *McKenzie, Collier & Waycott, 2015*), dugong numbers had declined abruptly (*Marsh et al., 2001*; *Sobtzick et al., 2012*), hawksbill turtles were in decline (*Bell, Schwarzkopf & Manicom, 2012*) and some sharks, rays, and large fish populations were in decline, especially in coastal and inshore areas (*Chin et al., 2010*; *Ceccarelli et al., 2014*). Overall, even before the 2016 bleaching event, the inherent resilience of this globally significant reef ecosystem had been seriously compromised and there was a concern that numerous elements of the biological diversity for which it is renowned may have been at risk (*World Heritage Committee, 2014*, *2015*; *Day, 2015*).

In complex and dynamic ecosystems like coral reefs, biodiversity in the true sense of the word—encompassing genetic, species, community, and ecosystem diversity is enigmatic (*Gaston, 1998*). On the GBR at least 12,000 species of marine vertebrates and invertebrates have been reported (Table 1; *Pitcher et al., 2007*; *Hutchings, Kingsford & Hoegh-Guldberg, 2008*). However, this estimate of species diversity is highly conservative. For many groups, only preliminary estimates are available and these are often based on survey data that were collected 20–40 years ago. Furthermore, condition and trend information is only available for only a limited number of species (*Fabricius & De'ath, 2001*; *GBRMPA, 2014a*); hence even for many large charismatic species of high conservation significance, the status of populations is uncertain (*Hamann & Chin, 2015*). Cryptic lineages (*Schmidt-Roach et al., 2013*); new species (e.g. *Hooper & Van Soest, 2006*; *Miller & Downie, 2009*; *Sutcliffe, Hooper & Pitcher, 2010*; *Hunter & Cribb, 2012*; *Schmidt-Roach, Miller & Andreakis, 2013*; *Capa & Murray, 2015*) and even entire habitats such as sponge gardens, mesophotic reefs, and deep water corals, are still being discovered and mapped (e.g. *Bridge & Guinotte, 2012*; *Bridge et al., 2012*; *Beaman, 2012*; *Harris et al., 2013*; *López-Cabrera et al., 2016*; *McNeil et al., 2016*).

**Table 1 Threatened species on the Great Barrier Reef.**

| | Number of species recorded on the GBR* | Listed threatened species | |
| --- | --- | --- | --- |
| | | Australia's environment protection and biodiversity conservation act 1999 | Global red list index (critically endangered, endangered or vulnerable) |
| Sponges# | 2,500 | 0 | Not assessed |
| Jellyfish# | 100 | 0 | Not assessed |
| Soft corals and sea pens# | 150 | 0 | Not assessed |
| Ascidians/tunicates# | 720 | 0 | Not assessed |
| Bryozoans# | 950 | 0 | Not assessed |
| Anemones# | 40 | 0 | Not assessed |
| Hard corals# | 450 | 0 | 88 |
| Echinoderms# | 630 | 0 | 10 |
| Crustaceans# | 1,300 | 0 | Not assessed |
| Molluscs# | 3,000 | 0 | 2 |
| Insects and arachnids# | 25 | 0 | Not assessed |
| Worms# | 500 | 0 | Not assessed |
| Bony fishes | 1,625 | 1 | 5 |
| Sharks and rays∞ | 136 | 9 | 21 |
| Breeding sea snakes | 14 | 0 | 0 |
| Marine turtles | 6 | 6 | 5 |
| Whales and dolphins | 30 | 6 | 4 |
| Dugong | 1 | 1 | 1 |
| **Total** | **12,177** | **23** | **136** |

Notes:
The number of animal species known to occur on the GBR contrasted with the number listed on national and global threatened species lists. Adapted from the 2014 Outlook Report (GBRMPA, 2014a) and the 2016 IUCN Red List.
* Excludes crocodile, nesting seabirds, shorebirds, plankton and marine flora.
# Best available estimate.
∞ A. Chin, 2016, unpublished data.

## Current approaches to informing coral reef management for biodiversity conservation

The task of protecting marine biodiversity is immense and exacerbated by the logistic challenges of conducting species-level surveys and the high level of taxonomic expertise needed to identify species. Hence, including a large number of species in routine monitoring can be regarded as both impractical and in some cases ineffective (Bottrill et al., 2008). Moreover, in contemporary conservation science there is a growing view that monitoring can be a waste of resources rather than a prerequisite for optimal management (Legg & Nagy, 2006). This perspective has been fuelled by the decades of monitoring studies that have reported population declines with no apparent link to management objectives (Nichols & Williams, 2006), or without any responsive action being taken. Furthermore, the amount of money and capacity required to protect all biodiversity is considered astronomical and far beyond the current investment in conservation action (James, Gaston & Balmford, 2001). Hence, by necessity, ecosystem management approaches adopt a triage approach that involves prioritizing the investment

of scarce resources in a reduced set of factors that are more manageable (*Bottrill et al., 2008*; *Wilson et al., 2011*).

The principal way managers are informed about the status of biodiversity is through surrogate information (defined in *Hunter et al., 2016*). In some cases, surrogate information can relate to subsets of data about indicator species (*Gardner et al., 2008*), cross-taxon surrogates (*Rodrigues & Brooks, 2007*), broad habitat-based proxy metrics (*Dalleau et al., 2010*) or abiotic surrogates (*Beier et al., 2015*). Some have argued that proxies reduce the time and cost required for data collection (*Humphries, Williams & Vane-Wright, 1995*; *Favareau et al., 2006*) suggesting that developing indicator, surrogate or proxy metrics that adequately represent diversity trends is an important and pragmatic conservation objective (*Baillie et al., 2008*). Numerous other studies, however, have questioned the ability for proxy metrics to effectively represent diversity (*Araújo et al., 2001*; *Rodrigues & Brooks, 2007*; *Andleman & Fagan, 2000*), highlighting that all proxy metrics have limitations (*Pressey, 2004*) especially if their performance is not evaluated with empirical data (*Vellend, Lilley & Starzomski, 2008*). One study examining the efficacy of biological surrogacy in seabed assemblages on the GBR indicated that no one taxonomic group was a particularly good surrogate for another and recommended that examining multiple taxonomic groups together was the preferred approach (*Sutcliffe et al., 2012*).

On the GBR, the overwhelming majority of diversity surrogate information is collected (and reported) at a habitat level. Relatively little data is available at the species-level. For example, all but two of the references cited in the 2014 Strategic Assessment (*GBRMPA, 2014b*) and the 2014 Outlook Report (*GBRMPA, 2014a*) to substantiate the condition and trend of coral species diversity on the GBR relate to the percentage of live coral cover. Likewise, a recent paper documenting the impact of the 2016 coral bleaching event (*Hughes et al., 2017*) reports species-level responses for only two of the 410 coral species known to occur on the GBR despite claiming to document the resistant corals and susceptible species. Moreover, 21 taxa were recorded at the level of genera, two genera were further broken down into growth forms, three taxa were reported at the family level and soft corals were reported at the level of order. Despite their important role as ecosystem engineers, for hard corals, there has not been a comprehensive species-level assessment of coral diversity on the GBR since 2001 (*DeVantier et al., 2006*) and species-level information is not routinely collected or reported on, hence the only species-level data available to underpin management decisions about the status of coral biodiversity on the GBR is now over 15 years old.

The situation is worse for other neglected marine taxa—no current information on status or trends exists for the overwhelming majority of marine taxa including highly targeted and vulnerable taxa such as sea cucumbers and giant clams (*Purcell et al., 2014*). Thus, given multiple disturbances (e.g. including bleaching and mortality events, cyclones, freshwater flood plumes, and a new outbreak of crown of thorn (COT) sea stars (*Acanthaster planci*) have impacted the GBR over the last decade, the questions remain—how can we be sure we are adequately protecting biodiversity if we have only a partial understanding of what is there? Do we have adequate

data to evaluate how biodiversity has responded to both management efforts and disturbance events?

## Coral cover is a poor surrogate for biodiversity

On coral reefs, habitat proxies are commonly used to quantify the condition of coral reefs, with percent live hard coral cover being the most widely used metric in monitoring studies (*Bruno & Selig, 2007*; *Eakin et al., 2010*; *De'ath et al., 2012*). It is stated in the Reef 2050 Plan that the extent, condition and trend of habitat provide the best indicators of biodiversity (*Commonwealth of Australia, 2015*). However, despite its broad use, hard coral cover is not a robust indicator of coral diversity. A study undertaken at Lizard Island on the GBR showed that coral cover was not an effective proxy for coral diversity because coral cover is not related to species richness as a positive linear function (*Richards, 2013*). To further explore the generality of this finding, the study was repeated at four additional locations (Ashmore Reef, Kosrae-Micronesia, Majuro-Marshall Is., and Christmas Island; *Richards & Hobbs, 2014*; *Ryan, Richards & Hobbs, 2014*). In those studies, percent live coral cover consistently performed poorly as an indicator of coral species richness (Fig. 2).

Numerous studies have maintained that the level of coral cover can be used to make inferences about fish biomass (*Chabanet et al., 1997*); the prevalence of coral diseases (*Bruno et al., 2007*); reef accretion potential (*Perry et al., 2012*) and arguably reef aesthetics (*Pocock, 2002*) or the economic value of reefs relating to tourism (*Stoeckl et al., 2011*). However, focusing solely on collecting data on habitat condition using general indicators such as coral cover is counter-productive to species conservation because when used in isolation, habitat proxies are not informative about ecological condition (*Hughes et al., 2010*), functionality (*Alvarez-Filip et al., 2013*) or diversity (*Richards, 2013*). Moreover, the finding of high coral cover can be deceptive because cover can be driven by the dominance of a small number of species whilst a high level of diversity can be sustained even when coral cover is low or moderate. Thus, it is important to note that a habitat with high coral cover does not guarantee functional diversity (defined by *Cadotte, 2011* as the trait variation in an assemblage), or community reassembly after disturbance (*Hughes et al., 2010*). Communities with high coral cover may be dominated by one or a small number of species and a single disturbance event (i.e. selective bleaching, disease outbreak or COTs infestation) could wipe out these dominant habitat formers with cascading ecosystem effects. With greater diversity comes the increased likelihood that some species would survive disturbance events and that critical functions such as reef building will be maintained. Thus, monitoring coral cover alone can fail to alert decision-makers to declines in resilience and it may in-fact mask losses of resilience.

## The risk of silent extinctions in the absence of an effective understanding of biodiversity

Without appropriate empirical baseline data (e.g. current population size and trend data), it is impossible to accurately detect or predict population growth, depletions, range shifts, to identify species with superior tolerance or to understand how species

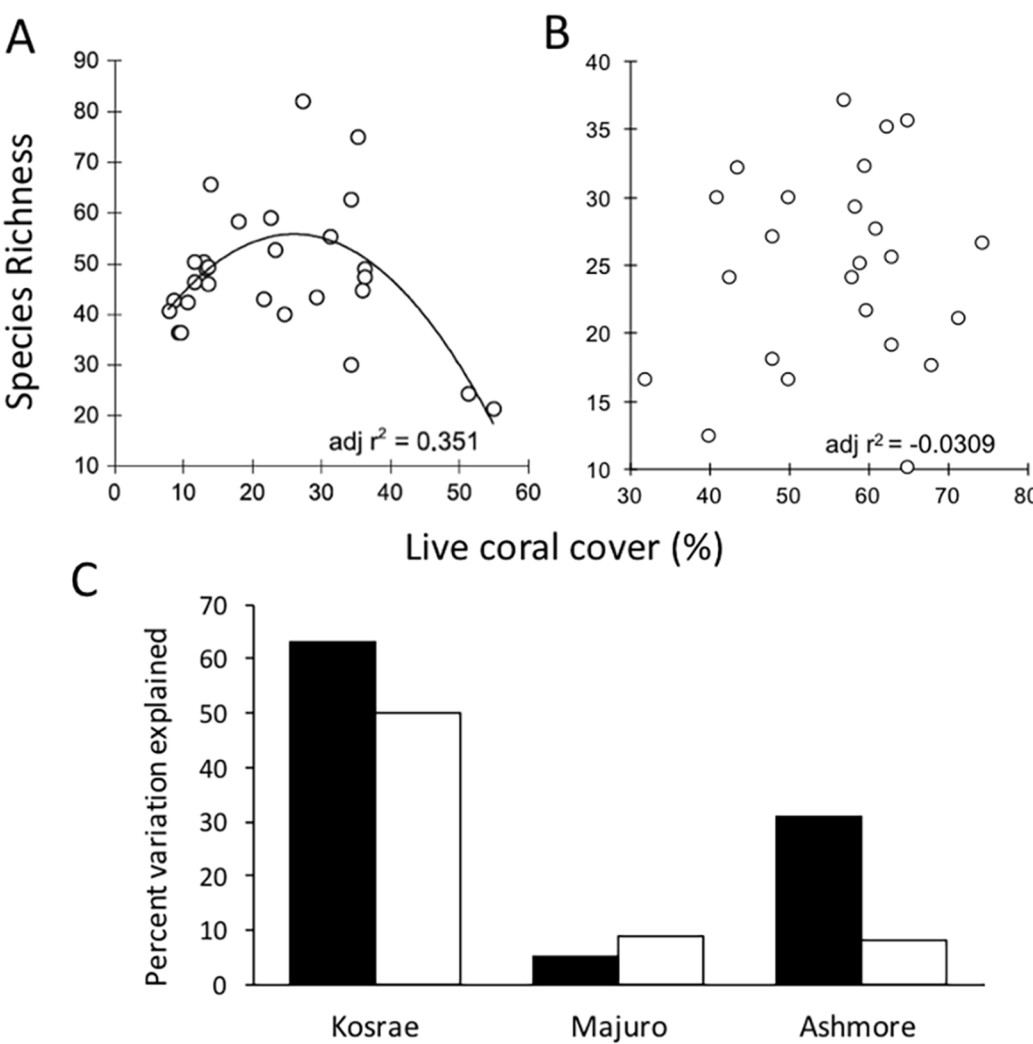

**Figure 2 The relationship between mean coral species richness and mean percent live coral cover.** (A). The results of a survey conducted at Lizard Island based on 28 sites showed that there was a polynomial relationship between coral species richness and percent live coral cover (adj $r^2$ = 0.351, SE 10.789, df = 25, $p$ = 0.001) and significance of the quadratic term in multiple regression suggested a non-linear relationship where species richness peaked at intermediate levels of coral cover (Adapted from *Richards, 2013*). (B). When repeated at Christmas Island, no significant ($p > 0.05$) linear or polynomial relationships were found between percent live coral cover and species richness at any scale (site, depth or transect) (Adapted from *Ryan, Richards & Hobbs, 2014*). (C). When the relationship was analysed at three additional locations Pacific and Indian Ocean locations, the overall $r^2$ values were considered too low (i.e. below 0.7) to provide meaningful estimates of coral species richness regardless of the scale of analysis (black bars: site means; white bars: transect level) (Adapted from *Richards & Hobbs, 2014*).

respond to management effort. Furthermore, without baseline data it is likely that extinctions may be occurring as unrecorded *silent extinctions* (*Myers & Ottensmeyer, 2005*). Silent extinctions may initially take place as ecological extinctions (when a species is reduced to such low abundance that, although still present, it no longer plays its typical ecological role); but they may also occur as undetected local extinctions

(the disappearance of a species from part of its range) (*Estes, Duggins & Rathbun, 1989*). Both of these types of extinctions are precursors to wider extinction events.

Even though there have been few, if any, known extinctions on the GBR, there is reason to believe that the threat of extinction is growing in tandem with the increasing frequency and velocity of disturbance events. The GBR Strategic Assessment (*GBRMPA, 2014b*) states there are significant concerns about a small number of species, including the spear tooth shark which may now be extinct in the GBR Region. While some potentially at risk species on the GBR are protected by other legislative mechanisms (e.g. all species of sawfish which are protected under Queensland legislation, *GBRMPA, 2013b*)—others are not. For example, two species of sea cucumbers (*Holothuria whitmaei* and *H. fuscogilva*) that are listed as *Endangered* and *Vulnerable* on the IUCN red-list index (RLI) (*IUCN, 2015*) have been heavily fished, and the annual catch of another Vulnerable species (*Stichopus herrmanni*) rose at an average annual pace of 200% from 2007 to 2011 (*Eriksson & Byrne, 2013*) however none of these species are protected under Australian legislation.

Other habitat-forming species such as hard corals are listed as no-take species under the GBRMPA Act 1975 which affords them protection from extractive threats. However, no stony coral species are specifically listed as threatened species under the *EPBC Act*, and their actual population status is not known nor is the effectiveness of current legislation as a regulatory control. Furthermore, with no post-impact data recorded at a species-level, it is impossible to report the impact that disturbance events (such as the recent widespread coral bleaching event) have had, nor is it possible to substantiate species recovery.

The significant and escalating situation on the GBR was placed in the international spotlight by UNESCO's World Heritage Committee (WHC) in 2011, and the Committee has since continued to raise concerns about the status of the GBR World Heritage Area (e.g. *World Heritage Committee, 2013*, *2014*, *2015*). In 2014, the WHC requested that the Australian government provide concrete and consistent management measures to ensure the overall long-term conservation of the property, including addressing cumulative impacts and increase reef resilience. In response, the Reef 2050 Plan was drafted (*Commonwealth of Australia, 2015*). As highlighted elsewhere (*Australian Academy of Science, 2015*; *Hughes, Day & Brodie, 2015*; *Roth et al., 2017*), the Reef 2050 Plan falls short on responding to these requests especially in the context of addressing climate change, shipping impacts, setting realistic, and measurable targets, cumulative impacts and a commitment to the level of funding required. The Reef 2050 Plan does, however, provide some useful actions and targets to monitor population trends for some mega fauna such as dugongs and turtles. The provision of strategies to evaluate and report on the population trends of other non-charismatic taxa like benthic invertebrates is notably absent. For example, the condition and trend for the 'other invertebrates' section in the GBR Strategic Assessment is listed as 'very good' and 'stable' but there is no confidence in the data and very limited evidence (*GBRMPA, 2014b*). Also in the 'sea snakes' section of the same assessment, it states abundance estimates are only available for a few species or for small areas, and there is little information about population trends.

## Scale-based mismatches in threatened species assessments

For most of the 12,000+ animal species that are documented from the GBR (Table 1), regional status information is notably absent (exceptions are nesting seabirds and shorebirds). However, for approximately 700 species including hard corals, sea cucumbers, giant clams, grouper, wrasse, parrotfish, sharks, rays, sea snakes, turtles, whales, dolphins, and dugong (*Carpenter et al., 2008*; *Elfes et al., 2013*; *Sadovy de Mitcheson et al., 2013*; *Conand et al., 2014*; *Dulvy et al., 2014*), conservation status has been assessed at a global level using the Red List of Threatened Species (RLI) (*IUCN, 2015*). The global assessments indicate that at least 136 species on the GBR (approximately 20% of species assessed) occur in elevated categories of threat (*Critically Endangered, Endangered* or *Vulnerable*) (Table 1) including at least 89 species of hard coral, 10 species of sea cucumbers, 21 species of sharks and rays, two species of giant clam, and five species of bony fish (Table S2).

Despite the wider background level of threat for these 136 species, only 23 of them (17%) are listed as threatened under the *Environment Protection and Biodiversity Conservation Act, 1999* (*Australian Government, 2015*) or defined as protected species under the *Great Barrier Reef Marine Park Regulations, 1983*. The regional status of the other 113 assessed species that have been recognised as highly threatened on a global scale by their respective taxonomic experts is not known (e.g. the Narrow Sawfish (*A. cuspidata*), Purple Eagle Ray (*Myliobatus hamlyni*), Humphead Wrasse (*Cheilinus undulatus*, see Fig. 3) and five species of sea cucumbers (Holothridae)). Nor is the status of an additional 35 species that are listed on the RLI as Data Deficient (Table S2, only three of which are protected under EPBC), or the other 11,000+ animal species known or likely to occur on the GBR which have not been assessed at any scale (global, national or regional, see Table 1). These species are not protected under the *Environment Protection and Biodiversity Conservation Act, 1999* because they are not endemic to Australia and the global IUCN assessments have little bearing on decisions made under the EPBC Act unless the species in question is endemic to Australia.

## Future directions

In order to effectively manage the GBR a multi-pronged approach is needed (*McCook et al., 2009*). Pragmatism has led to a preference for system-wide ecosystem management with the result that species-level initiatives have been de-emphasised. However, obtaining a comprehensive and up-to-date understanding of the status of GBR marine life through targeted taxonomic and ecological research on key knowledge gaps and long-term species monitoring remains a fundamental way to make informed management decisions (*GBRMPA, 2009*, *2013a*, *2014a*) and creates opportunities to engage with the public and bolster support for biodiversity conservation. The key argument against species-level surveys is that they are prohibitively expensive. But is this true? By way of example, a large multi-institutional, multi-taxon marine biodiversity project (*Marine Life of Kimberley* project) was recently conducted in NW Australia. Just like the GBR, the Kimberley marine bioregion comprises diverse coastal and offshore habitats and a large number of islands with fringing reef habitats. However, it covers a larger and

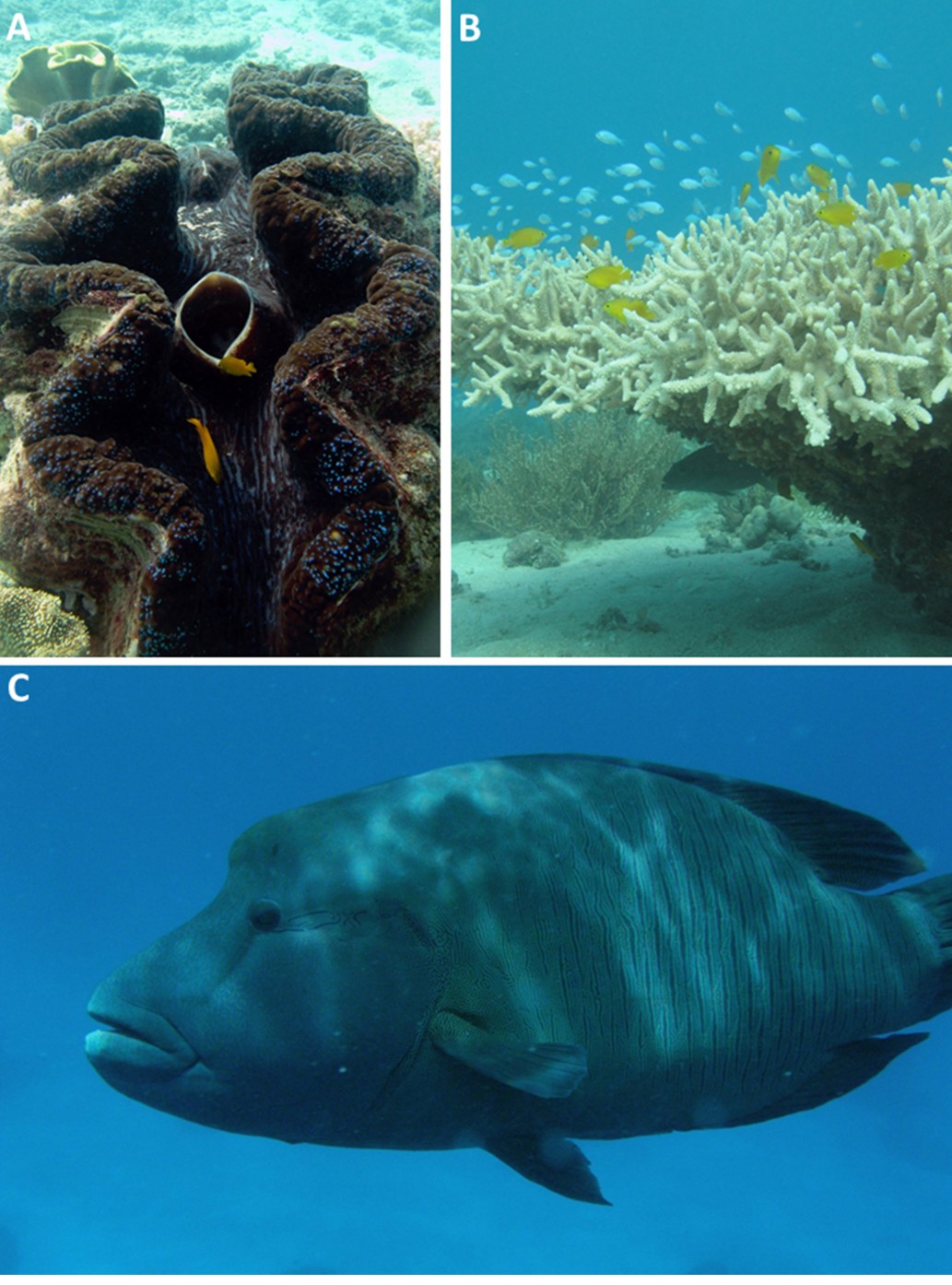

**Figure 3 Examples of species that are under threat at a global scale but their regional status is not known.** (A) *Tridacna gigas*, a vulnerable clam species (see http://www.iucnredlist.org/details/22137/0). (B) *Acropora donei*, a vulnerable staghorn coral species (see http://www.iucnredlist.org/details/133223/0). (C) *Cheilinus undulatus*, an endangered species of wrasse (see http://www.iucnredlist.org/details/4592/0). Photos by Zoe Richards.               

even more remote area (Kimberley project area is 476,000 km² (*Bryce, Bryce & Radford, 2018*); the GBR is 347,800 km² (*GBRMPA, 2014a*)). The *Kimberley Marine Life* project had two phases. The first phase (2008–2011) involved the compilation and analyses of

historic data from several Australian museums and the WA Herbarium to give a state of current knowledge for the region's marine life. Phase 2 (2011–2015) involved a series of surveys to increase the resolution of the marine diversity data and fill knowledge gaps identified during Phase 1. The project targeted the region's molluscs, crustaceans, fish, hard corals, soft corals, sponges, echinoderms, worms, marine algae, and sea grasses (*Bryce & Sampey, 2014*; *Fromont & Sampey, 2014*; *Hosie et al., 2015*; *Huisman & Sampey, 2014*; *Hutchings et al., 2014*; *Moore et al., 2015*; *Richards, Sampey & Marsh, 2014*; *Sampey & Marsh, 2015*; *Willan, Bryce & Slack-Smith, 2015*). This $2.7 AUD million dollar project (plus in-kind support) has provided a comprehensive reference dataset to guide in the design of the newly gazetted marine parks in the region; informed a range of research projects (WAMSI; http://www.wamsi.org.au); led to award-winning educational outreach opportunities (http://museum.wa.gov.au/btw/) and established a knowledge legacy for future generations. If an equivalent annual financial investment was made into updating our understanding of the status of marine life on the GBR, this would equate to less than 0.42% of the estimated annual worth of the reef to Australia's economy (valued at $6.4 billion annually, *Deloitte Access Economics, 2017*).

If new biodiversity studies are undertaken on the GBR, it would be crucial for this new information to be made available to decision-makers in a timely manner to ensure enduring and constructive links are made between monitoring and subsequent management actions (*Day, 2008*). More specifically, it must align with the five-year GBR outlook reporting schedule (the next report is due in 2019). In lieu of new surveys, a value of information analysis (*Runge, Converse & Lyons, 2011*; *Moore & Runge, 2012*; *Yokomizo, Coutts & Possingham, 2014*) could be undertaken to provide scientifically credible advice on how, when, and what new information is needed to broaden and accelerate efforts to conserve the biodiversity values of the GBR. Such an analysis may, for example, find there is a need to expand the existing monitoring program for the GBR to include key diversity indices. For some taxa, biodiversity proxies have been established e.g. particular families have been identified as good indicators of wider reef fish diversity (*Allen & Werner, 2002*), however for most, this information is not available and more research is needed in this area.

Any expansion of the existing GBR long-term monitoring program would need to be carefully considered and negotiated amongst a body of experts, managers, stakeholders, independent advisors, and preferably, informed by a value of information analysis. As a start, some possible options for adapting and expanding the monitoring program include:

1. Targeted top and tail monitoring which involves collating species-level data on a selection of the most common (keystone) species in addition to some of the most endangered and vulnerable species and/or invasive species.

2. Rotational monitoring whereby different taxa are monitored in a four-year cycle to increase the breadth of species captured in the overall monitoring program (see Fig. 4).

3. Stratified monitoring that involves in-depth taxonomic and ecological assessments at a smaller number of sites that overlap with key long-term monitoring stations.

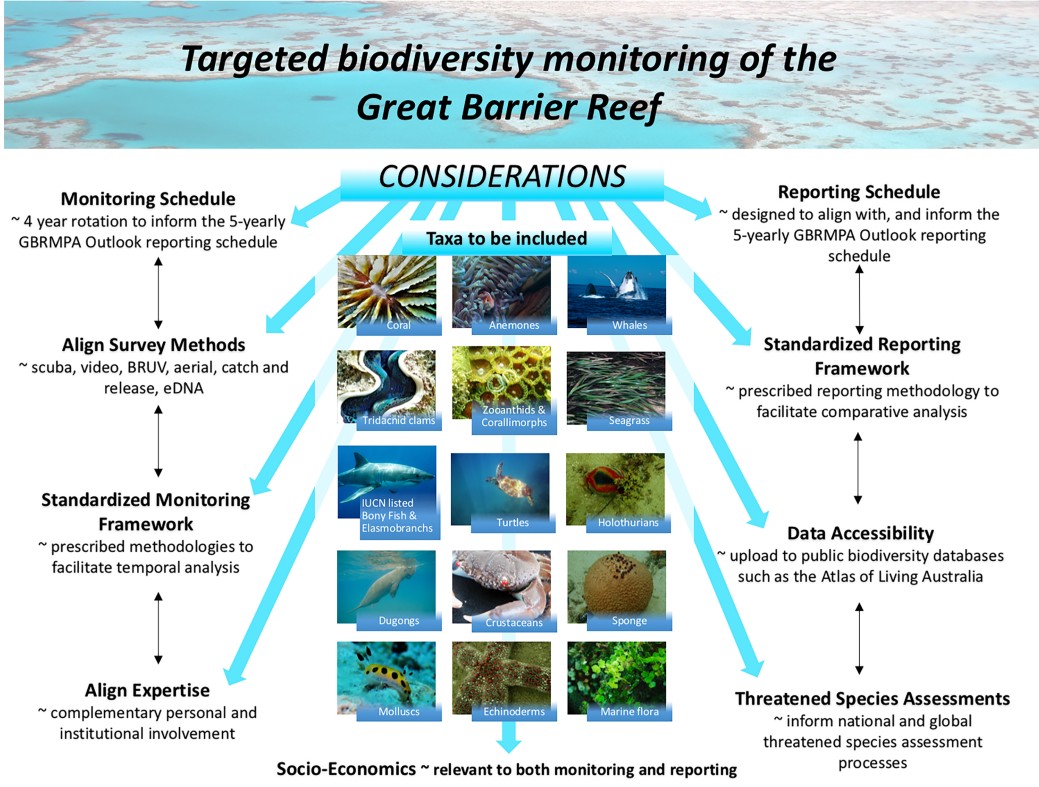

**Figure 4 Rotational biodiversity monitoring is one example of how existing monitoring programs could be adapted.** If long-term monitoring programs are adapted, all decisions relating to target species and yearly taxa subsets would require expert discussion following baseline surveys or a Value of Information Analysis. Such a program would require resourcing and co-ordination but the vision could be for it to be conducted by two to three scientists per year (rotating based on expertise) and undertaken alongside the existing long-term monitoring program. Credits: Aerial—Photographer: M. Cowlishaw, Copyright Commonwealth of Australia (GBRMPA); Humpback Whale—Photographer: M. Simmons, Copyright Commonwealth of Australia (GBRMPA); Sea grass—Photographer: J. Jones, Copyright Commonwealth of Australia (GBRMPA); Great White Shark—Photographer: K. Hoppen, Copyright Commonwealth of Australia (GBRMPA); *Halimeda*—Photographer: G. Goby, Copyright Commonwealth of Australia (GBRMPA); Dugong—Photographer: B. Cropp, Copyright Commonwealth of Australia (GBRMPA); Turtle—Photographer: E. Goodwin, Copyright Commonwealth of Australia (GBRMPA). All other images by Zoe Richards.

While these are just a few examples of possible ways the monitoring and reporting program could evolve, they illustrate that if preventing coral biodiversity loss is a priority to coral reef management authorities, it is essential to begin the process of collecting species-level data, using that information to choose appropriate indicators or surrogates (*Beier et al., 2015*) and adapting monitoring programs to collect additional complementary data that more effectively informs managers and the public about the status of biodiversity.

In addition, there are other reporting opportunities. In some cases, existing photographs and video records could be retrospectively analysed to identify trends for key species of interest (e.g. giant clams). Moreover, citizen science projects (such as ReefLife Surveys) hold promise for providing data on charismatic or easily identified

fauna. For some taxa, estimates of generic diversity (*Richards, 2013*, *Richards & Hobbs, 2014*) or functional diversity (*Cadotte, 2011*; *Cernansky, 2017*) may be useful and pragmatic additions to monitoring programs. An additional promising approach is to audit species composition is through the application of environmental DNA (eDNA) meta-barcoding technologies. Advances in DNA sequencing technologies and the swift drop in the cost of sequencing has led to a rapid rise in the applications of eDNA, particularly in the marine environment (e.g. *Foote et al., 2012*; *Lejzerowicz et al., 2015*; *Thomsen et al., 2012*; *Valentini et al., 2016*; *Miya et al., 2015*). The power and limitations surrounding eDNA as a tool for marine surveys of diversity, diet, food webs and invasive species is currently being explored by numerous molecular laboratories. So far it appears that so long as careful attention is paid to meta-barcoding workflows, assay development and the development of taxonomically sound reference datasets, eDNA represents a powerful new tool. eDNA is increasingly being used to evaluate the composition and health of marine communities (*Foote et al., 2012*; *Murray, Coghlan & Bunce, 2015*; *Clarke et al., 2017*) including coral reefs (*Stat et al., 2017*; *Shinzato et al., 2018*) and may be a useful complement to traditional monitoring approaches.

## CONCLUSION

Most of the threats identified to be harming the diversity of marine life on the GBR over the last two–three decades remains to be effectively addressed and many are worsening. The habitat degradation that has already occurred will take decades to reverse and far greater resources than are currently being expended are required to document and protect biodiversity on the GBR (*Day, 2015*; *Brodie, 2016*). What is urgently required is decisive and effective conservation action by key leaders to ensure the legal obligations of protecting biodiversity are met. Establishing sound taxonomic datasets and ensuring biodiversity is adequately and representatively monitored are urgent prerequisites for achieving efficient conservation plans and mitigating biodiversity loss (*Wilson, 2016*; *Troudet et al., 2017*; *Thomson et al., 2018*). The problem of having a limited understanding of the species that inhibit our oceans and the threats they face is not restricted to the GBR or Australia (*Webb & Mindel, 2015*). However, for Australia, the problem could also be an opportunity for best-practice marine conservation. The changing nature of the GBR necessitates we consider what additional dynamic approaches to monitoring and reporting are needed to more fully understand the current status of biodiversity and mitigate the risk that silent extinction events will occur. Unless more action is taken to evaluate and manage biodiversity and to provide current and comprehensive assessments of extinction risk (*Costello, May & Stork, 2013*; *Bland et al., 2015*; *Costello, 2015*), we may fail to pass onto future generations many of the values that comprise what is universally regarded as the world's most iconic coral reef ecosystem.

### Funding

This work was supported by the WA Museum and Woodside Energy and a Curtin Research Fellowship awarded to Zoe T. Richards. The work was undertaken as part of

Linkage Project LP160101508. The funders had no role in study design, data collection and analysis, decision to publish, or preparation of the manuscript.

### Grant Disclosures
The following grant information was disclosed by the authors:
WA Museum and Woodside Energy and a Curtin Research Fellowship.
Linkage Project: LP160101508.

### Competing Interests
The authors declare that they have no competing interests.

### Author Contributions
- Zoe T. Richards conceived and designed the experiments, analysed the data, prepared figures and/or tables, authored or reviewed drafts of the paper, approved the final draft.
- Jon C. Day prepared figures and/or tables, authored or reviewed drafts of the paper, approved the final draft.

### Data Availability
This research did not generate any data or code.

### Supplemental Information
Supplemental information for this article can be found online at http://dx.doi.org/10.7717/peerj.4747#supplemental-information.

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
