# Peer review of "Biodiversity of the Great Barrier Reef—how adequately is it protected?"

_PeerJ, doi:10.7717/peerj.4747_

## Round 0.1 · original submission · Major Revisions

I have heard back from two reviewers, and apologize for the delay; it took some time over the winter break to find suitable people. Both reviewers (and myself) feel this is a well-timed and needed piece of work. The reviewers offer constructive suggestions to improve your work, and while I do not think any new analyses are needed, some more explanation on your methods and perhaps some reorganization and focus on coral reefs are needed. Thus, while my decision is 'major revisions', you can consider this on the lesser side of major. I look forward to seeing your revised version.

Reviewer 1 ·

Basic reporting

This manuscript is presented as a literature review around the theme of biodiversity of marine communities along the Great Barrier Reef (GBR). The authors have attempted to collate existing literature which reports on 1) the status of biodiversity, 2) processes threatening species diversity and 3) the extent of existing biodiversity monitoring and reporting on the GBR.
I found the basic premise of the study to be a worthwhile focus, i.e. highlighting the paucity of data at the species level for many communities and the need to rectify this given the expanding disturbance regimes that are affecting not only the GBR but coral reefs around the globe.
The inference of the paper is that it addresses all marine communities but the vast majority of the manuscript is about corals, which clearly reflects the expertise of the lead author. Hence I believe this study would be better if it explicitly focussed on coral communities to highlight in detail the issues the authors are referring to. By all means keep the tables which highlight issues across all taxa, which readers could use as a starting point into the literature on other communities but the detail necessary to drive home the points that the authors are trying to make is to be found in the coral literature. I could even suggest some interesting wordplay to generate more interest in the topic e.g. along the lines of ‘Don’t forget the trees while trying to save the forests’.
The general writing style of the paper is unfortunately verbose and reads for the most part like a pep talk. For example the word biodiversity is mentioned 69 times in the main document – 82 times if the Abstract is included. I know that Biodiversity is the main theme of the paper but once the theme is established and framed in the Introduction then please refrain from literally shouting the term biodiversity at the reader every paragraph thereafter. There is also too much verbatim quoting from documents which continues to feel like the reader is being harangued about the importance/relevance of the authors study.

Experimental design

There are very clear methods around undertaking literature searches and I found the description of the survey methodology to be very unclear and very scattergun in approach. The question to be asked is “could I replicate the exact search process that the authors undertook?” I don’t believe I could, based on the description given. There needs to be a clear tabulation of themes, search terms used and search engines used. Then a summary of how the literature was then reviewed and the final documentation decided on.

Validity of the findings

As mentioned in my comments under basic reporting I found the manuscript to be far too emotive and repetitive in its messages to extract clear and concise outcomes from the work.

Additional comments

I found many statements that were unclear.
Line 87 – What is meant by biodiversity values being one of a complex array of multiple use values?
Line 98 - ……articles relating direct or indirect threats relating to the….(repeating of words – also seen in line 152, and again in lines 284 and 288 where the phrase ‘falls short’ is repeated)
Line 117 – Results section starts here
Line 131 – I don’t see the GBR as recovering from a legacy of impacts on crocodiles and whales? Populations of both of these taxa are healthy.
Lines 134-138 – more info on what is meant by habitats and species. % declines are stated but without baseline info included. E.g. coral cover has declined by 0.5% year – is this for the past 20 years or the past 50 years? Likewise for calcification rates.
Lines 162-164 – incomplete sentence and hence unclear – use the phrase ‘not only’ and then do not provide the subject this refers to.
Line 165 – Couldn’t find reference for Wilson (2007) in list.
Lines 172-175 I do not understand this sentence at all?
Line 323 – the comment “…where there is a large amount of uncertainty” needs clarification. The GBR is a well-managed system by national and international standards and the continued use of emotive terms without clarification continues the tone of this manuscript as a pep-talk.
Lines 327 and 330 – repeating phrase…”up-to-date understanding of biodiversity on GBR”
The conclusion – this section does not add anything to the document other than continue the emotive phrasing and entreaties of the rest of the manuscript.
Table S1 – in the section on Unsustainable fishing impacts, removal of herbivorous fish is mentioned as an unsustainable impact on the GBR. This is not as issue because herbivorous fishes are not targeted by Australian fishers, either commercial or recreational.

Quantifying, monitoring and managing biodiversity is extremely important but your review needs to be less a lecture about how important biodiversity is and more a clear examination - using corals – of how species level information can provide a much clearer picture of community dynamics in the face of disturbances. Proxies for biodiversity is a topic that needs much more work and this is a particularly relevant area to investigate – as you do – given the constraints with resources and funding that are the reality of field-based science. There is a fish paper that you have not discovered that provides an example of how surveys of the main fish families can be used to great effect to estimate total species richness of fish in a region. This would provide another example of how to get more value from surveys w.r.t biodiversity information. See Allen & Werner (2002) Env. Biol. Fishes 65:209-214.

Reviewer 2 ·

Basic reporting

Captured under general comments for the authors

Experimental design

Captured under general comments for the authors

Validity of the findings

Captured under general comments for the authors

Additional comments

Review of Richards and Day.

General comments.
This is a timely review. Great Barrier Reef Marine Park Authority and partners are progressing the Reef Integrated Monitoring and Reporting Program to support the Reef 2050 Plan. The ideas and recommendations presented by the authors of this paper will be relevant to the discussion of an effective program design.

The text can be tightened in a few places and I provide some specific recommendations below.

Line 37: World Heritage Status more due to outstanding universal value, of which biodiversity is an element. The Coral Triangle has higher biodiversity than the GBR.
L 43-46: This is much needed
L 50 – 55: Agree, but I’m not sure more monitoring data will safeguard species. Recommendations for effective management could be based on existing data and models
L 55-61: Great initiative, but again, monitoring may not be the limiting factor. In a time of more coral bleaching and other growing climate impacts (acidification, storms), the loss of habitat will mean less scope for biodiversity. Monitoring the system more intensely will require resources (taken form management budgets) and may not provide solutions. Safeguarding key habitats could.

L 61-62: Agree on the action, but I’m not sure more monitoring of “species of interest” is really effective action. What I’d say we need is a Value of Information analysis that helps us understand what new and targeted information we need to better manage the system to safeguard species and ecosystem values.
L 82-84. This needs a reference. I’d contest the idea that biodiversity buffers environmental variation. It would be more correct to say that biodiversity potentially comes with functional redundancy, which can increase resilience (Nystroem etc).
L 88: Should read “Fernandez”
L 119-124. Tighten the language here and sharpen the impact categories. The word “unsustainable” is loaded unless you explain it. Is “detrimentally” a word?
L 124: Use “extensive” instead of “pervasive”
L 125-133. Again, the text is loaded with adjectives. The far north has so far been spared from the land-use run-off issues of the wet tropics, but I don’t think the far north was ever considered pristine or a regional refuge. Rather than referring to “unsustainable” activities (which needs to be demonstrated to be called unsustainable), instead use “cumulative pressures”. You can use Anthony 2016 (Coral reefs under climate change and ocean acidification - challenges and opportunities for management and policy. Annual Review of Environment and Resources 41:59–81) as a reference.

L 161-174. This section is spot-on. However, I was disappointed to see that the authors don’t follow up on these conclusions and recommend ways for targeted/focused monitoring to link with management strategies. Instead, they leave the thread hanging and move on to talk about surrogates.
L 182: Surrogates and proxies are just that – they cannot “accurately represent trends in biodiversity”.
L 192-205. As you point out yourself in L161-174, monitoring is a trade-off challenge, so reporting changes at the level of species for coral reefs that may have a million species (Fisher, Knowlton etc) is infeasible and could paralyse large-scale monitoring programs. However, deeper taxonomic resolution at a few key sites could provide insight into biodiversity dynamics.
L 206-2013. Again, a Value of Information assessment is needed. Up-scaled monitoring to measure everything may not be the most effective way to manage a system in stress and under limited resourcing. Protecting key habitats and understanding risks to sensitive species that are likely to represent the responses of other species groups could be a cost-effective way to safeguard biodiversity without measuring everything.
L 206-247. Agree coral cover is insufficient as a stand-alone indicator. All good points here that the science and management community need to hear (again).

L 247-294. Excellent section. A new report that provides recommendations for the Reef 2050 review options is now available and could be used as a reference here. http://www.environment.gov.au/system/files/resources/3ae3225f-eb5c-418b-8889-e523a4fbb4e5/files/reef2050-plan-review-options-final-report.pdf
L 323-324. Explain what you mean by uncertainty.
L 330-331. Biodiversity assessments could still be part of a pragmatic monitoring and management approach if stratified so that in-depth assessments are made at a smaller number of sites that overlap with key long-term monitoring stations.
L 354-367. Good idea. Such a program could also be complemented with citizen-science (i.e. crowd) analysis of detailed online photographs from monitoring stations. This means the world could essentially help monitor the GBR.

---

## Round 0.2 · Minor Revisions

I am very sorry for the delay; we had expected one reviewer to provide comments but this did not materialize. I have gone over the paper myself instead, and find it to be very much improved. I have only a few small comments and one concern to be addressed, which I detail below. Thus, my decision is "minor revisions", with the emphasis on minor.

Major concern

1. I still think the methodology of the literature search could be a bit more clear. In Table S1, this includes (I think) all 124 references; it might be good to explicitly state this. Also, how were these references selected? What was the process? You describe it in some detail, but how did you (for example) select some references from the 2014 GBR Outlook Report (GBRMPA 2014) and not others? Or from the JCU library etc? Did you aim for a breadth of taxonomic coverage? I guess what I am trying to say is, just as a previous reviewer mentioned, I am not sure given the current explanation I could replicate all of your choices, and a bit more logic is needed in your explanation. Please contact me if you are uncertain by what I mean.

Minor comments

1. Word (marked up version) line 112: remove comma after "Hughes". Also line 155 for similar unneeded comma after author's name - and numerous other locations after this.
2. lines 360-361 - give scientific name of COTS here please.
3. lines 455-456 - remove the hyphens, and isolate the species name with commas on each side ", XXXX, "
4. I think the reference on line 748 is not quite complete; please check. Other references (Dulvy et al. 2003, etc) need some formatting to be consistent, watch for capitalization and commas, periods, journal abbreviations, etc.
5. line 1066 - Halimeda needs to be italicized.

---

## Round 0.3 · accepted · Accept

Thank you for taking care of these last edits and revisions; the methodology of the literature review is now much clearer. The paper is now ready to be published - I look forward to seeing the final version!

#